# GUIDING TRANSFORMERS TO PROCESS IN STEPS

## ABSTRACT

Neural networks have matched or surpassed human abilities in many tasks that humans solve quickly and unconsciously, i.e., via Kahneman's "System 1", but have not been as successful when applied to "System 2" tasks that involve conscious multi-step reasoning. In this work, we argue that the kind of training that works for System 1 tasks is not sufficient for System 2 tasks, propose an alternative, and empirically demonstrate its effectiveness. Specifically, while learning a direct mapping from inputs to outputs is feasible for System 1 tasks, we argue that algorithmic System 2 tasks can only be solved by learning a mapping from inputs to outputs through a series of intermediate steps. We first show that by using enough intermediate steps a 1-layer 1-head Transformer can in principle compute any finite function, proving the generality of the approach. We then show empirically that a 1-layer 1-head Transformer cannot learn to compute the sum of binary numbers directly from the inputs, but is able to compute the sum when trained to first generate a series of intermediate results. This demonstrates, at a small scale, how a fixed-size neural network can lack the expressivity to encode the direct input-output mapping for an algorithmic task and yet be fully capable of computing the outputs through intermediate steps. Finally, we show that a Frozen Pretrained Transformer is able to learn binary addition when trained to compute the carry bits before the sum, while it fails to learn the task without using intermediates. These results indicate that explicitly guiding the neural networks through the intermediate computations can be an effective approach for tackling algorithmic tasks.

## 1 INTRODUCTION

Daniel Kahneman has pointed out that there is a fundamental difference in how humans solve the following two tasks (Kahneman, 2011):

(1) complete the phrase "bread and ...";

(2) complete the equation "$17 \times 24 = \ldots$".

The answer to (1) comes to mind instantly, with no mental effort, and is a result of a computation that one is not consciously aware of and could not explain. The answer to (2) requires time and effort to come up with, and is a result of a sequence of computations that one consciously carries out. The former mode of cognition is what Kahneman calls System 1, the latter System 2.

We currently have separate tools for solving each of these two kinds of problems on a computer, but it is not yet clear how to build a single system capable of both modes of cognition simultaneously. Many tasks that exercise System 2 in humans involve executing some fully-specified algorithm and thus are quite straightforward to solve using conventional computer programs. However, classical programming is not applicable to System 1 tasks as these typically correspond to functions that we do not know how to implement. Instead, System 1 tasks are usually tackled by approximating the target functions from observed input-output pairs. Whether such a learning-based approach could be successfully extended to System 2 tasks is an open question, and one that we aim to explore in this paper.

We argue that the classical System 1 paradigm of training neural networks to approximate functions from inputs and outputs corresponding to a given task is not a scalable approach for tackling tasks from the System 2 domain. We claim that, given only inputs and outputs, neural networks will

not receive enough supervision to converge to a solution when trained to perform algorithmic tasks that humans solve via System 2. To circumvent this, we propose to supplement the training data with intermediate results that would be helpful to compute before arriving at the final output. By modifying the training data in this way, we are effectively changing the training objective — instead of having to learn any arbitrary way of mapping inputs to outputs, the neural networks are now trained to compute the outputs from the inputs through a particular sequence of intermediate steps. We hypothesize that such additional guidance might be necessary in order for neural networks to learn algorithmic tasks, and we empirically evaluate this hypothesis.

Much of existing work in applying neural networks for System 2 tasks has been focused on making neural architectures more aligned with the execution of algorithms. While fine-tuning the model architectures may ultimately lead to superior System 2 capabilities (after all, the architecture of a calculator allows it to perform arithmetic at a super-human level), we instead explore the challenge of extending the capabilities of existing state-of-the-art System 1 models — namely, the Transformer — towards the System 2 domain. In particular, we take inspiration from the human ability to solve algorithmic tasks by writing down symbols on paper and propose to formulate such tasks as sequence-to-sequence problems where the target sequence includes both the outputs and all the intermediate symbols that a human would write (or more). That is, instead of modifying the models, we experiment with modifying the data. Our main contributions can be summarized as follows:

- We show that it is possible to hand-code a 1-layer 1-head Transformer to compute any finite function if enough intermediate steps are used;

- We show that a 1-layer 1-head Transformer can be trained to perform binary addition if the target sequences include intermediate results, whereas it fails to learn the task when predicting the outputs directly from the inputs; and

- We show that a Frozen Pretrained Transformer can be trained to perform binary addition if the target sequences include intermediate results, whereas it fails to learn the task when predicting the outputs directly from the inputs.

## 2 RELATED WORK

The need of bridging the gap between System 1 and System 2 capabilities in current deep learning models has recently been emphasized by Yoshua Bengio (Bengio, 2019; Bengio et al., 2021). The work of Bengio and colleagues addresses the higher-level cognition in its broadest sense, seeking to model and incorporate into current systems concepts such as consciousness (Bengio, 2017), causality (Bengio et al., 2020), agency (Thomas et al., 2018), and global workspace (Goyal et al., 2021). In Goyal & Bengio (2020), they question the paradigm of classical statistical learning and propose to shift from training models on curated datasets towards training agents in complex non-stationary environments. In contrast to that, our approach lies within a conventional learning framework (namely, sequence-to-sequence modeling with Transformers), and addresses System 2 in a slightly narrower sense (namely, as algorithmic execution).

A considerable fraction of recent work in applying neural networks for algorithmic tasks has been concerned with modifying or augmenting neural architectures to make them more suitable for algorithmic execution. One very common approach involves trying to bring components from classical computer architecture into the neural setting: Neural Turing Machines give neural networks access to dynamic external memory (Graves et al., 2014; 2016), Stack-augmented RNNs augment the networks with an infinite pushdown stack (Joulin & Mikolov, 2015), Neural Random Access Machines use registers and introduce the ability to manipulate and dereference pointers (Kurach et al., 2016), while Neural Arithmetic Logic Modules represent neural versions of the ALU (Trask et al., 2018; Madsen & Johansen, 2020). A parallel but closely related line of research frames these new kinds of augmented architectures as neural controllers endowed with access to external interfaces and applies reinforcement learning techniques to train them to solve algorithmic tasks by interacting with the interfaces (Zaremba & Sutskever, 2015; Zaremba et al., 2016).

The approach of using variable amount of computation per input, known as Adaptive Computation Time (ACT), has been introduced by Graves (2016). Universal Transformers (Dehghani et al., 2019) integrate ACT into the Transformer architecture, allowing each output symbol to be a result of a variable number of applications of a single Transformer layer. While these works are, like ours,

motivated by the apparent necessity of intermediate processing in certain tasks, they use *learned* ponder time and *learned* ponder content whereas our work uses *given* ponder time and *given* ponder content. The main advantage of learned ponder time and content is that the model is free to discover and learn to perform the intermediate computations that it finds most useful. This also means that the researcher does not need to know the correct intermediate steps to train the model and can thus tackle a potentially larger class of problems. The main disadvantage of learned ponder time and content is that the amount of supervision per forward pass decreases as the model uses more intermediate steps, and the signal can quickly become too weak for the model to train at all. It is also worth noting that intermediate steps are typically given when humans are taught to perform algorithmic tasks (rather than being asked to infer intermediate computations by just looking at input-output pairs), which might be an important argument for given ponder content.

The idea of providing supervision beyond input-output examples has already appeared in several projects, although, in our view, it still remains largely underexplored. Reed & de Freitas (2016) train Neural Programmer-Interpreters by providing supervision on the correct action sequences (execution traces) of the recurrent controller, Mirman et al. (2018) train differential Neural Computational Machines with extra supervision on the movements of the read-write heads, and Mirman et al. (2018) train Neural Execution Engines with extra supervision on the attention masks. While the added training signal does lead to better sample efficiency and improved generalization, a major limitation of all of these approaches, as acknowledged by Mirman et al. (2018), is that the extra supervision needs to be highly specialized as it applies to very specific components of each architecture. Our approach of supplementing the target sequences with intermediate results is completely architecture-independent and thus does not share this limitation. Veličković et al. (2020) use extra supervision at the data level as well, though their work is focused specifically on tasks involving graph-structured inputs and is thus based on neural network architectures that are tailored for processing graphs. In contrast to that, we adopt a general sequence-to-sequence modeling framework, with an intention to imitate a human executing an algorithm on a piece of paper. Our main goal is to explore how and whether powerful existing System 1 models could be used in the System 2 domain, rather than to find a neural architecture that would be most suitable for executing algorithms. We therefore use vanilla decoder-only Transformers in our experiments.

## 3 MOTIVATION

The distinguishing characteristic of System 1 tasks is that they can be solved instantly and unconsciously, in something akin to a single "forward pass" through a human neural network. It thus seems plausible that all input-output mappings corresponding to System 1 tasks are in principle computable via a single forward pass through a sufficiently large artificial neural network. Recent success in deep learning shows that training bigger models on more data indeed makes it possible to solve more and more System 1 tasks, and it is not unreasonable to expect this trend to continue.

When it comes to solving System 2 tasks, however, scaling up appears to be a dead-end. Consider the problem of completing the sentence "The first digit of the $n$-th Fibonacci number is ...", where $n$ is replaced by some positive integer. Since a single forward pass through a neural network (no matter how large) involves a constant amount of computation, there will be some number $N \in \mathbb{N}$ such that for all $n > N$ the correct output is not computable. It follows that the only viable approach for solving these kinds of algorithmic tasks is to arrive at the output through intermediate steps.

The main value of introducing intermediate steps is that it gives control over the level of model expressivity required to implement a solution to a particular task. No matter how many atomic operations separate the outputs from the inputs in a given algorithmic problem, performing those exact operations in sequence would make each execution step only as complex as a single atomic operation. This is arguably the main reason why a human brain can "implement" something like integer multiplication even though it does not contain a circuit for multiplying arbitrarily large numbers directly — the key is that a complex problem can be decomposed into simpler ones until each sub-problem becomes solvable "atomically" via System 1. By the same token, a neural network that is not expressive enough to compute some output through 0 intermediate steps may be expressive enough to compute the same output through $l$ intermediate steps for some $l > 0$, since each of the $l$ steps would involve computing a simpler function. Since many System 2 tasks are algorithmic, their outputs can be made arbitrarily distant from the inputs (in terms of the number of atomic operations

separating the two), which implies that step-by-step processing is the only scalable approach for solving them.

The necessity of intermediate processing does not by itself imply the amount of guidance that the neural networks should receive during training. The exact intermediate computations can either remain unspecified and be inferred by the model, or be fixed and provided as part of the training data. Although there are pros and cons to both alternatives (as we have outlined in section 2), we see the latter as more promising and choose to include the intermediate results in the target sequences used during training. Our position is based on the following three observations: first, it is natural to provide complete supervision over the intermediate computations when teaching humans to perform algorithmic tasks; second, learned intermediates create the vanishing supervision problem (the longer the ponder time, the less training signal is received per forward pass); and third, for all algorithmic tasks the exact intermediate results are fully known anyway. While manually increasing the amount of supervision goes against the tradition of minimizing the reliance on expert knowledge that arguably lead to most of the recent achievements in the System 1 domain, System 2 tasks represent a structurally different class of problems for which a different set of constraints may apply.

The arguments laid out above have the following important implications:

- *Expressiveness*: a neural network implementing only some very basic atomic operations should in principle be capable of expressing arbitrarily complex functions as long as enough intermediate steps are used;
- *Training*: a neural network that fails to learn some algorithmic task when trained without intermediate processing may be able to learn the same task if enough intermediate steps are used; and
- *Composability*: a neural network that is pretrained on some general-purpose task and does not contain a circuit implementing some algorithmic computation directly may nevertheless be able to solve the same task by computing the output through simpler operations which it does contain circuits for.

We establish the first of these implications in Section 4 and we empirically validate the latter two in Sections 5 and 6. Even though demonstrating a single example where a statement holds does not prove it in general, our empirical results confirm that the hypothesized phenomena do occur in the particular contexts we considered in this work and thus serve as evidence in favor of the latter two implications.

## 4 EXPRESSIVENESS

Human beings are able to solve algorithmic problems, such as integer multiplication, by writing down sequences of symbols, usually with outputs at the end, where each new symbol is a result of some basic operation applied to some of the symbols preceding it. Intuitively, it seems that the Transformer architecture should have just the right ingredients for emulating this kind of processing: it is typically used for generating sequences of symbols by iteratively predicting the next one, and its attention mechanism allows each predicted symbol to be a function of selected preceding ones. We prove this intuition by demonstrating that it is possible to construct a 1-layer 1-head Transformer to compute any finite function through intermediate steps by emulating NOR circuits. The full details are included in Appendix A, but the general argument goes as follows:

(1) Since NOR is a universal gate, for every finite function $f : \{0,1\}^n \to \{0,1\}^m$ there is a Boolean circuit made up of NOR gates that computes $f$.

(2) By sorting the gates of such a NOR circuit according to their topological order, we can represent the circuit as a sequence of binary symbols $(s_1, \ldots, s_{n+l+m})$, where $(s_1, \ldots, s_n)$ are the inputs, $(s_{n+l+1}, \ldots, s_{n+l+m})$ are the outputs, and, for each $i > n$, $s_i = \text{NOR}(s_{j_i}, s_{k_i})$ with $j_i, k_i < i$.

(3) Given $(s_1, \ldots, s_n)$, we can correctly complete the sequence with a 1-layer 1-head decoder-only Transformer if its attention matrix $A$ satisfies $A_{i,z} = 0.5$ if $z \in \{j_i, k_i\}$ and $A_{i,z} = 0$ otherwise (the model attends to the correct pair of symbols at each step) and the position-wise feed-forward module computes the NOR of the two symbols attended to.

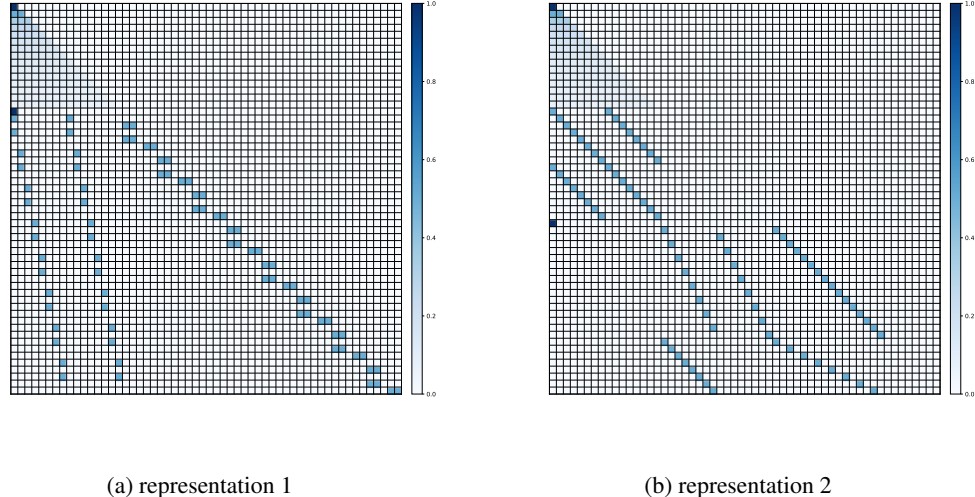

(a) representation 1                            (b) representation 2

Figure 1: Attention matrices from the hand-coded Transformer implementing binary addition with $n = 8$.

We present a concrete setting of all of the weights of a 1-layer 1-head decoder-only Transformer that makes (3) possible in Appendix A, confirming the first implication from Section 3.

While implementing NOR gates is sufficient for the proof due to the universality of the NOR gate, a slight modification to the weights of the feed-forward module allows us to choose between NOR, AND, and XOR operations at each position in the output sequence. As a concrete example, we used this construction to implement addition of fixed-length bit strings (with guaranteed perfect accuracy), proving that addition, with the right number of intermediates, is in the space of programs computable by a 1-layer 1-head Transformer. In particular, for each position $i$ along the inputs, letting $a_i$ and $b_i$ denote the input bits and $c_i$ denote the carry bit (obtained from the previous position, i.e., from the sum $a_{i-1} + b_{i-1} + c_{i-1}$), we compute

$$d_i = \text{XOR}(a_i, b_i)$$
$$e_i = \text{XOR}(c_i, d_i)$$
$$f_i = \text{AND}(a_i, b_i)$$
$$g_i = \text{AND}(c_i, d_i)$$
$$c_{i+1} = \text{XOR}(f_i, g_i)$$

where $e_i$ is the $i$-th bit of the sum and $c_{i+1}$ is the carry bit for the next position (defining $c_1 = 0$). We can thus represent the addition of, say, 3-bit numbers $a_1 a_2 a_3$ and $b_1 b_2 b_3$ (in little-endian order, where $a_1$ is the least significant bit) as the sequence

(representation 1)        $a_1 a_2 a_3 b_1 b_2 b_3 c_1 d_1 e_1 f_1 g_2 c_2 d_2 e_2 f_2 g_2 c_3 d_3 e_3 f_3 g_3 c_4,$

or, by reordering the symbols to put the sum at the end and intermediates in between the inputs and the outputs,

(representation 2)        $\underbrace{a_1 a_2 a_3 b_1 b_2 b_3}_{\text{inputs}} \underbrace{d_1 d_2 d_3 f_1 f_2 f_3 c_1 g_1 c_2 g_2 c_3 g_3 c_4}_{\text{intermediates}} \underbrace{e_1 e_2 e_3}_{\text{outputs}},$

which is the representation that we actually use in our training experiments. Figure 1 shows the attention matrices corresponding to both representations (row $i$ of the attention matrix indicates which symbols are attended to when predicting the $i$-th symbol from the output sequence). As can be verified from the figure, the hand-coded Transformer does indeed implement the correct attention patterns corresponding to two different representations of the Boolean circuit for binary addition.

## 5 TRAINING

One of our core hypotheses in this work is that in many System 2 tasks the outputs will be too many atomic operations away from the inputs to be computable in a single forward pass, regardless of how much the neural networks are scaled up, and that introducing intermediate computation is one way of solving the issue. If true, this phenomenon should manifest at small scale as well: a shallow model may lack the expressivity to learn a direct mapping from inputs to outputs for a relatively simple task but be expressive enough to learn the mapping from inputs to outputs through intermediates, as decomposing the problem would reduce the minimum required complexity of a single operation to the point where each individual step becomes computable in a single forward pass. We evaluate this hypothesis empirically by training a 1-layer 1-head decoder-only Transformer on the task of binary addition. (We use the minGPT implementation for our experiments (Karpathy, 2020).)

We chose to investigate binary addition primarily because of its simplicity. Conveniently, there are very clear candidates for what intermediate computations to use – e.g., we can supplement the training data with the carry bits (which are usually explicitly computed when a human is adding numbers on a piece of paper), or the values of all non-output gates in a Boolean circuit implementing binary addition (which is what we did in a previous section and do here as well). This means that, even if the task is decomposed into the most elementary operations (i.e., binary Boolean gates), each output symbol would be associated with at most $4$ intermediate computations and so the gap between the inputs and outputs that is left for storing the intermediate results will not be disproportionately large.

We chose to investigate the 1-layer 1-head Transformer for three main reasons. First, we know from Section 4 that this model is sufficient for emulating the Boolean circuit for binary addition and we know what the target attention patterns look like (Figure 1). Second, using only a single layer and a single attention head enhances the interpretability of the model as this makes it possible to unambiguously determine its attention patterns (there is only a single attention matrix we need to look at). Finally, recall that the goal of our experiment is to test whether a model that is unable to learn a task without the use of intermediate computations could learn the same task if intermediates were used. Thus, even though binary addition may well be a simple enough problem for a large transformer to solve without any intermediate steps at all, our aim here is to artificially restrict the expressivity of the transformer to a point where it is unable to learn the task without using intermediate steps so that we could then add intermediate computations and test whether this makes the task learnable. As the results of this section demonstrate, reducing both the number of layers and the number of heads of the Transformer to $1$ is sufficient to achieve this.

We frame binary addition as a task of sequence completion — that is, the model receives a prefix of $2n$ symbols representing the two input numbers and has to predict the subsequent $l + n$ symbols, where $l$ is the number of intermediate symbols and the last $n$ symbols represent the sum (ignoring the overflow bit). We train the model in 3 different regimes:

(1) *Weak supervision*: the model is trained to predict the sum immediately after the inputs, i.e., $l = 0$. A single training example for $n = 3$ would have the form $a_1a_2a_3b_1b_2b_3e_1e_2e_3$, with loss computed on the model's predictions of $e_1e_2e_3$.

(2) *Medium supervision*: the model is trained to predict the sum after $l = 4n+1$ intermediates, with loss computed only on the sum and not the intermediates. A single training example for $n = 3$ would have the form $a_1a_2a_3b_1b_2b_3d_1d_2d_3f_1f_2f_3c_1g_1c_2g_2c_3g_3c_4e_1e_2e_3$, with loss computed on the model's predictions of $e_1e_2e_3$.

(3) *Strong supervision*: the model is trained to predict both the sum and the $l = 4n + 1$ intermediates obtained by decomposing binary addition down to logic gates as explained in section 3. A single training example for $n = 3$ would have the form $a_1a_2a_3b_1b_2b_3d_1d_2d_3f_1f_2f_3c_1g_1c_2g_2c_3g_3c_4e_1e_2e_3$, with loss computed on the model's predictions of $d_1d_2d_3f_1f_2f_3c_1g_1c_2g_2c_3g_3c_4e_1e_2e_3$.

In all three regimes we use teacher-forcing during training, meaning that when predicting the $i$-th symbol of the sequence the model sees the correct target symbols at all positions up to $i$. During testing, the model receives only the two input numbers (e.g., $a_1a_2a_3b_1b_2b_3$) and is used to generate the completion (e.g., $e_1e_2e_3$ for weak supervision and $x_1x_2x_3x_4x_5x_6x_7x_8x_9x_{10}x_{11}x_{12}x_{13}e_1e_2e_3$ for medium and strong supervision) autoregressively using greedy sampling.

Table 1: Test set accuracy of a 1-layer 1-head Transformer on binary addition of up to $n$-bit numbers.

| | $n = 8$ | $n = 16$ | $n = 24$ | $n = 32$ | $n = 40$ | $n = 48$ |
|---|---|---|---|---|---|---|
| weak supervision | 1.0000 | 0.2529 | 0.1773 | 0.1961 | 0.0720 | 0.0000 |
| medium supervision | 0.0473 | 0.0003 | 0.0000 | 0.0000 | 0.0000 | 0.0000 |
| strong supervision | 1.0000 | 1.0000 | 1.0000 | 1.0000 | 0.8916 | 0.9873 |

We run experiments with $n \in \{8, 16, 24, 32, 40, 48\}$. For each $n$, we perform a grid search over $5 \times 2 \times 3 \times 3 = 90$ hyperparameter settings and run 4 randomly initialized trials for each of those. The hyperparameters we vary are the embedding dimension (chosen from $\{32, 64, 128, 256, 512\}$), the number of training examples (chosen from $\{1000, 10000\}$), the positional encodings (either fixed and defined via the sine and cosine functions as in the original Transformer (Vaswani et al., 2017), fixed and set to the hand-coded values as in Appendix A, or learned), and the training regime (weak, medium, or strong supervision). In Table 1, we report the maximum accuracy achieved with weak, medium, and strong supervision over all hyperparameter settings and random trials for each $n$. We measure the exact-match accuracy against the target output bits representing the sum of the input numbers. The test set consists of 10000 examples not seen by the model during training.

The results in Table 1 show that, for $n \geq 16$, a 1-layer 1-head Transformer is not able to learn binary addition from weak or medium supervision while it can learn to solve the problem when strong supervision is used, supporting the second implication from Section 3. This experiment thus serves as a demonstration of a concrete problem for which the use of intermediate steps is the key factor determining whether a neural network will be able to solve a task or not. This supports our main hypothesis that constant-size neural networks incapable of encoding the input-output mappings corresponding to algorithmic tasks may be capable of solving these same tasks by encoding the mappings from inputs to outputs through a series of intermediate steps.

We also include a visualization of the attention patterns of a Transformer trained with strong supervision in Figure 2 (specifically, we plot the attention matrix averaged over 100 evaluations on random inputs), along with the target patterns taken from the hand-coded model. As indicated by the considerable resemblance between most of the corresponding rows from the two plots, the trained Transformer does indeed learn to compute the intermediate results correctly and make use of them when predicting the outputs. This confirms that guiding Transformers through particular chains of intermediate computations can be instrumental in training them to perform algorithmic tasks.

Finally, for more context on the impact of hyperparameters on performance, Figures 3 and 4 compare the maximum test accuracies achieved with different positional encodings and sizes of the training dataset, respectively. We find that the best performance is achieved by using the sinusoidal positional encodings as used in the original Transformer (Vaswani et al., 2017), and that the smaller dataset of 1000 examples is sufficient to achieve the best performance in most cases (the only exception being the strong supervision case with $n = 40$).

## 6 COMPOSABILITY

Lu et al. (2021) have demonstrated that the weights of large pretrained Transformers encode general-purpose computations which can be leveraged to solve tasks from a diverse set of modalities. They show that it is possible to achieve competitive accuracies on a range of tasks by fine-tuning only the input layer, the output layer, and the layer-norm parameters of a pretrained GPT-2 model while freezing all of the remaining weights in the "body" of the model (which comprise more than 99.9% of the parameters). This restricted version of the model is called a Frozen Pretrained Transformer (FPT).

While it is demonstrated that FPT is able to achieve perfect accuracy on symbol manipulation tasks like bit memory or bitwise XOR, these tasks only involve elementary single-step computations and thus fall into the domain of System 1. The question we are interested in is whether the weights of a pretrained language model contain circuits for solving non-trivial multi-step algorithmic tasks, and if not, whether solutions to such tasks can be assembled by composing simpler circuits that are

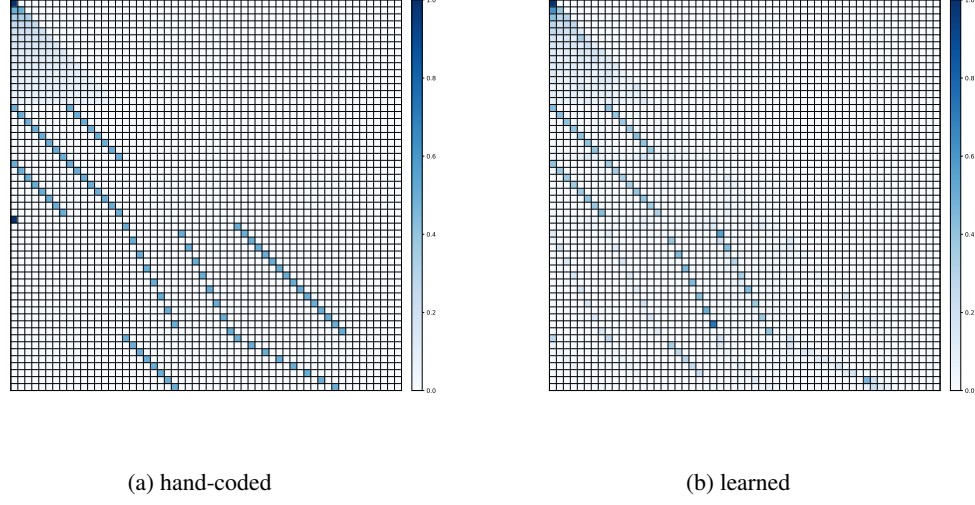

(a) hand-coded            (b) learned

Figure 2: Attention matrices from the hand-coded vs. trained Transformer implementing binary addition with $n = 8$.

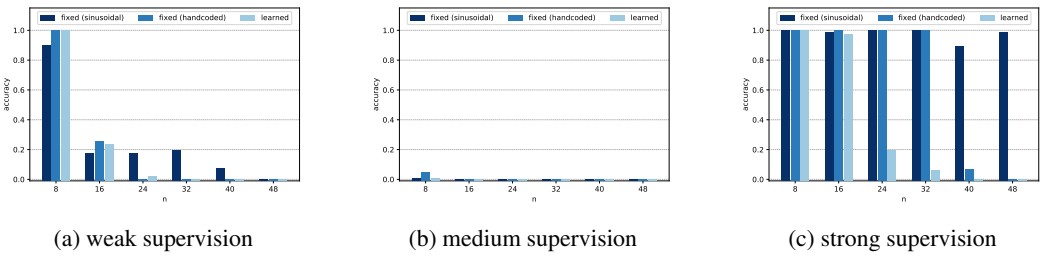

(a) weak supervision      (b) medium supervision      (c) strong supervision

Figure 3: Test set accuracy of a 1-layer 1-head Transformer on binary addition of up to $n$-bit numbers with different positional encodings.

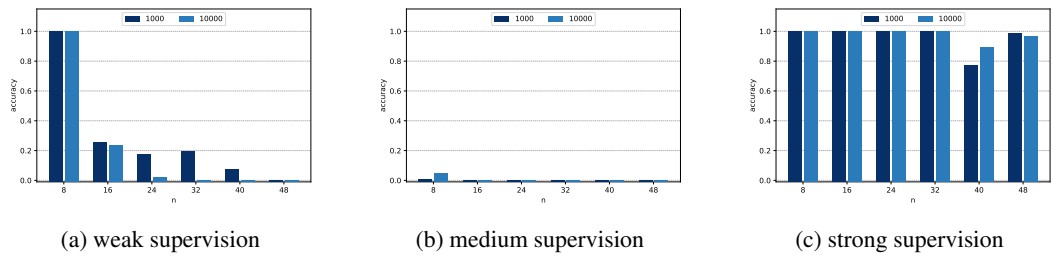

(a) weak supervision      (b) medium supervision      (c) strong supervision

Figure 4: Test set accuracy of a 1-layer 1-head Transformer on binary addition of up to $n$-bit numbers with different training set sizes.

Table 2: Test set accuracy of a Frozen Pretrained Transformer on binary addition of up to $n$-bit numbers.

|  | $n = 8$ | $n = 16$ | $n = 32$ | $n = 64$ | $n = 128$ |
|---|---|---|---|---|---|
| weak supervision | 1.0000 | 0.0005 | 0.0000 | 0.0000 | 0.0000 |
| medium supervision | 0.0357 | 0.0000 | 0.0000 | 0.0000 | 0.0000 |
| strong supervision | 1.0000 | 1.0000 | 1.0000 | 1.0000 | 1.0000 |

contained within the weights. If the weights could encode a direct input-output mapping, fine-tuning the FPT on example inputs and outputs would be sufficient for priming the model to perform the task. However, if the solution could only be composed out of pieces, we would expect FPT to learn the task only when fine-tuned to produce enough intermediate results before the output such that the individual computations are expressible via the pretrained weights.

We investigate this by training a FPT on binary addition in the weak, medium, and strong supervision regimes analogous to those described in Section 5, though, in contrast to the previous experiments, we only use the carry bits as the intermediates (FPT is a full-size multi-layer Transformer and thus may require fewer intermediate steps than a 1-layer 1-head model). For instance, with $n = 3$, FPT would receive $a_1 a_2 a_3 b_1 b_2 b_3$ as input and have to predict $e_1 e_2 e_3$ in the weak supervision regime and $c_2 c_3 c_4 e_1 e_2 e_3$ in the strong supervision regime (using the same notation as before). We train on 10000 examples for up to 100 epochs using teacher forcing and test on 10000 unseen examples without teacher forcing. We report the exact match accuracies (against the sum bits) over the test examples in Table 2. As we can see, FPT fails to learn binary addition in the weak and medium supervision regimes while it is able to solve the task when being trained to compute the carry bits first. This suggests that binary addition may not be directly implemented in the weights of a pretrained language model but can nevertheless be expressed as a combination of more granular operations, supporting the third implication from Section 3. This result seems to be an analog of a similar feature of the System 2 learning capabilities of humans — while it may not be possible to teach a child to compute each digit of the sum in one unconscious step, the same child can easily learn to compute the correct outputs by explicitly calculating and keeping track of the carries. Once again, we see that guidance over intermediate computations is what a neural network's ability to solve an algorithmic task appears to hinge on.

## 7 CONCLUSION

In this work, we have argued that the only way for fixed-size neural networks to solve algorithmic System 2 tasks is by computing the outputs through intermediate steps. If neural networks are going to be trained to perform such tasks from data, the actual intermediate results will either be given or have to be inferred. We here focused on the regime with given intermediates, called strong supervision. Our experiments demonstrated that a 1-layer 1-head Transformer and a Frozen Pretrained Transformer can learn to perform binary addition only if strong supervision is used, supporting our hypothesis that guidance over intermediate computations can be necessary for solving algorithmic tasks.

We believe that strong supervision will be an integral part of any scalable attempt to use a learning-based approach for tackling algorithmic tasks from the domain of System 2. We speculate that the usefulness of strong supervision might even extend beyond the space of algorithmic tasks, since any cognitive task that is characterized by distinct periods of thought leading to concrete intermediate results would become easier to learn if examples of those intermediate results were given during training. This leads us to believe that the idea of providing explicit guidance over intermediate computations might point to fruitful directions for future research.

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

## A    APPENDIX

A 1-layer 1-head decoder-only Transformer is a parameterized function mapping sequences of symbols to next symbol probabilities, with parameters given by

$$
\begin{aligned}
\theta = \{&W_{\text{emb}}, W_{\text{pos}}\} \\
\cup \{&\gamma_{\text{ln1}}, \beta_{\text{ln1}}, W_{\text{query}}, b_{\text{query}}, W_{\text{key}}, b_{\text{key}}, W_{\text{value}}, b_{\text{value}}\} \\
\cup \{&\gamma_{\text{ln2}}, \beta_{\text{ln2}}, W_{\text{ff1}}, b_{\text{ff1}}, W_{\text{ff2}}, b_{\text{ff2}}\} \\
\cup \{&\gamma_{\text{ln3}}, \beta_{\text{ln3}}, W_{\text{out}}\}.
\end{aligned}
$$

Given an input sequence of $b$ symbols $(s_1, s_2, \ldots, s_b)$ from some vocabulary $\mathcal{V}$, the Transformer outputs $b$ vectors $h_{\text{prob}}^{(i)} \in [0, 1]^{|\mathcal{V}|}$ for $i \in \{1, 2, \ldots, b\}$ interpreted as conditional PMFs for next symbol prediction and computed via the following definitions:

$$h_{\text{emb}}^{(i)} = (W_{\text{emb}})_{s_i+1,:} + (W_{\text{pos}})_{i,:} \tag{1}$$

$$h_{\text{ln1}}^{(i)} = \text{LayerNorm}_{\gamma_{\text{ln1}}, \beta_{\text{ln1}}}\left(h_{\text{emb}}^{(i)}\right) \tag{2}$$

$$q^{(i)} = h_{\text{ln1}}^{(i)} W_{\text{query}} + b_{\text{query}} \tag{3}$$

$$k^{(i)} = h_{\text{ln1}}^{(i)} W_{\text{key}} + b_{\text{key}} \tag{4}$$

$$v^{(i)} = h_{\text{ln1}}^{(i)} W_{\text{value}} + b_{\text{value}} \tag{5}$$

$$K_{i,:} = k^{(i)} \tag{6}$$

$$a^{(i)} = \text{Softmax}\left(\frac{q^{(i)} K^\top}{\sqrt{d}} + \text{Mask}_{i,:}\right) \tag{7}$$

$$h_{\text{att}}^{(i)} = \sum_{j=1}^{b} a_j^{(i)} v^{(j)} \tag{8}$$

$$h_{\text{res1}}^{(i)} = h_{\text{att}}^{(i)} + h_{\text{ln1}}^{(i)} \tag{9}$$

$$h_{\text{ln2}}^{(i)} = \text{LayerNorm}_{\gamma_{\text{ln2}}, \beta_{\text{ln2}}}\left(h_{\text{res1}}^{(i)}\right) \tag{10}$$

$$h_{\text{ff1}}^{(i)} = \text{ReLU}\left(h_{\text{ln2}}^{(i)} W_{\text{ff1}} + b_{\text{ff1}}\right) \tag{11}$$

$$h_{\text{ff2}}^{(i)} = \text{ReLU}\left(h_{\text{ff1}}^{(i)} W_{\text{ff2}} + b_{\text{ff2}}\right) \tag{12}$$

$$h_{\text{res2}}^{(i)} = h_{\text{ff2}}^{(i)} + h_{\text{res1}}^{(i)} \tag{13}$$

$$h_{\text{ln3}}^{(i)} = \text{LayerNorm}_{\gamma_{\text{ln3}}, \beta_{\text{ln3}}}\left(h_{\text{res2}}^{(i)}\right) \tag{14}$$

$$h_{\text{prob}}^{(i)} = \text{Softmax}\left(h_{\text{ln3}}^{(i)} W_{\text{out}}\right), \tag{15}$$

where $d$ is the embedding dimension, and Mask is an autoregressive attention mask, i.e., a $d \times d$ matrix with all elements above the main diagonal equal to $-\infty$ and all of the remaining elements equal to $0$. Predictions are made by interpreting the entries of $h_{\text{prob}}^{(i)}$ as next-symbol probabilities, that is,

$$P_\theta(s_{i+1} = \mathcal{V}_j) = \left(h_{\text{prob}}^{(i)}\right)_j, \tag{16}$$

for each $j \in \{1, 2, \ldots, |\mathcal{V}|\}$, where $\mathcal{V}_j$ represents the $j$-th symbol from the vocabulary.

Let $n$ and $m$ be some positive integers and let $(s_1, s_2, \ldots, s_{n+m})$ be a sequence of binary symbols where, for each $i > n$, $s_i = \text{NOR}(s_{j_i}, s_{k_i})$ for some $j_i, k_i < i$. We are going to show a concrete setting of the weights of a 1-layer 1-head decoder-only Transformer such that, given $(s_1, s_2, \ldots, s_n)$, it autoregressively predicts $(s_{n+1}, s_{n+2}, \ldots, s_{n+m})$ as the completion of the sequence. Specifically, given any $\epsilon^* > 0$, we can hand-code the parameters $\theta$ such that, for each $i \geq n$, $P_\theta(s_{i+1} = \text{NOR}(s_{j_{i+1}}, s_{k_{i+1}})) > 1 - \epsilon^*$.

Choose any $\epsilon^* > 0$. Let $b = n + m - 1$ be the block size, $d = 3 + b$ be the embedding dimension, and define constants

$$c_1 = 2 \left\lceil \ln \left( \frac{(i-2)(1/2 - \epsilon)}{2\epsilon} \right) \right\rceil, \quad c_2 = \left\lceil \ln \left( \frac{1 - \epsilon^*}{\epsilon^*} \right) \right\rceil, \quad \epsilon = \frac{1}{5 + 2d^2}, \quad \delta = 1 - 4\epsilon - 2d^2\epsilon.$$

Let the parameters of a Transformer be hand-coded according to the following definitions:

$$W_{\text{emb}} \in \mathbb{R}^{2 \times d} \quad \text{with} \quad (W_{\text{emb}})_{j,k} = \begin{cases} -1 & \text{if } (j,k) \in \{(1,1), (2,2), (1,3), (2,3)\} \\ 1 & \text{if } (j,k) \in \{(1,2), (2,1)\} \\ 0 & \text{otherwise,} \end{cases} \tag{17}$$

$$W_{\text{pos}} \in \mathbb{R}^{b \times d} \quad \text{with} \quad (W_{\text{pos}})_{j,k} = \begin{cases} 1 & \text{if } k = j + 3 \\ 0 & \text{otherwise,} \end{cases} \tag{18}$$

$$\gamma_{\text{ln1}} \in \mathbb{R}^d \quad \text{with} \quad (\gamma_{\text{ln1}})_j = 1, \tag{19}$$

$$\beta_{\text{ln1}} \in \mathbb{R}^d \quad \text{with} \quad (\beta_{\text{ln1}})_j = 0, \tag{20}$$

$$W_{\text{query}} \in \mathbb{R}^{d \times d} \quad \text{with} \quad (W_{\text{query}})_{j,k} = \begin{cases} (d/4)^{-1/2} c_1 & \text{if } k \in \{j_j, k_j\} \\ 0 & \text{otherwise,} \end{cases} \tag{21}$$

$$b_{\text{query}} \in \mathbb{R}^d \quad \text{with} \quad (b_{\text{query}})_j = 0 \tag{22}$$

$$W_{\text{key}} \in \mathbb{R}^{d \times d} \quad \text{with} \quad (W_{\text{key}})_{j,k} = \begin{cases} 1 & \text{if } j = k + 3 \\ 0 & \text{otherwise,} \end{cases} \tag{23}$$

$$b_{\text{key}} \in \mathbb{R}^d \quad \text{with} \quad (b_{\text{key}})_j = 0 \tag{24}$$

$$W_{\text{value}} \in \mathbb{R}^{d \times d} \quad \text{with} \quad (W_{\text{value}})_{j,k} = \begin{cases} (d/4)^{-1/2} & \text{if } (j,k) \in \{(1,4), (2,5)\} \\ 0 & \text{otherwise,} \end{cases} \tag{25}$$

$$b_{\text{value}} \in \mathbb{R}^d \quad \text{with} \quad (b_{\text{value}})_j = 0 \tag{26}$$

$$\gamma_{\text{ln2}} \in \mathbb{R}^d \quad \text{with} \quad (\gamma_{\text{ln2}})_j = 1, \tag{27}$$

$$\beta_{\text{ln2}} \in \mathbb{R}^d \quad \text{with} \quad (\beta_{\text{ln2}})_j = 0, \tag{28}$$

$$W_{\text{ff1}} \in \mathbb{R}^{d \times 4d} \quad \text{with} \quad (W_{\text{ff1}})_{j,k} = \begin{cases} -(d/4)^{1/2} & \text{if } (j,k) = (4,6) \\ (d/4)^{1/2} & \text{if } (j,k) = (4,7) \\ 0 & \text{otherwise,} \end{cases} \tag{29}$$

$$b_{\text{ff1}} \in \mathbb{R}^{4d} \quad \text{with} \quad (b_{\text{ff1}})_j = \begin{cases} -2d^2\epsilon & \text{if } j = 6 \\ 1 - 4\epsilon & \text{if } j = 7 \\ 0 & \text{otherwise,} \end{cases} \tag{30}$$

$$W_{\text{ff2}} \in \mathbb{R}^{4d \times d} \quad \text{with} \quad (W_{\text{ff2}})_{j,k} = \begin{cases} 1 & \text{if } (j,k) \in \{(6,6), (7,7)\} \\ -1 & \text{if } (j,k) \in \{(6,8), (7,9)\} \\ 0 & \text{otherwise,} \end{cases} \tag{31}$$

$$b_{\text{ff2}} \in \mathbb{R}^d \quad \text{with} \quad (b_{\text{ff2}})_j = 0 \tag{32}$$

$$\gamma_{\text{ln3}} \in \mathbb{R}^d \quad \text{with} \quad (\gamma_{\text{ln3}})_j = 1, \tag{33}$$

$$\beta_{\text{ln3}} \in \mathbb{R}^d \quad \text{with} \quad (\beta_{\text{ln3}})_j = 0, \tag{34}$$

$$W_{\text{out}} \in \mathbb{R}^{d \times 2} \quad \text{with} \quad (W_{\text{out}})_{j,k} = \begin{cases} \delta^{-1} d^2 c_2 & \text{if } (j,k) \in \{(7,1), (6,2)\} \\ 0 & \text{otherwise.} \end{cases} \tag{35}$$

Choose any $i \geq n$. After the token embedding and positional encoding layers, the hidden representation corresponding to symbol $i$ is given by

$$h_{\text{emb}}^{(i)} = (W_{\text{emb}})_{s_i + 1, :} + (W_{\text{pos}})_{i, :}. \tag{36}$$

Since

$$\left(h_{\text{ln1}}^{(i)}\right)_j = \text{LayerNorm}_{\gamma_{\text{ln1}},\beta_{\text{ln1}}}\left(h_{\text{emb}}^{(i)}\right)_j \tag{37}$$

$$= (\gamma_{\text{ln1}})_j \left(\frac{\left(h_{\text{emb}}^{(i)}\right)_j - \frac{1}{d}\sum_{k=1}^d \left(h_{\text{emb}}^{(i)}\right)_k}{\sqrt{\frac{1}{d}\sum_{k=1}^d \left(\left(h_{\text{emb}}^{(i)}\right)_k - \frac{1}{d}\sum_{k'=1}^d \left(h_{\text{emb}}^{(i)}\right)_{k'}\right)^2}}\right) + (\beta_{\text{ln1}})_j \tag{38}$$

$$= (d/4)^{1/2}\left(h_{\text{emb}}^{(i)}\right)_j, \tag{39}$$

after layer normalization the hidden representation is turned into

$$h_{\text{ln1}}^{(i)} = (d/4)^{1/2}h_{\text{emb}}^{(i)}. \tag{40}$$

Then, since the only nonzero elements of $h_{\text{ln1}}^{(j)}$ are $\left(h_{\text{ln1}}^{(j)}\right)_1$, $\left(h_{\text{ln1}}^{(j)}\right)_2$, $\left(h_{\text{ln1}}^{(j)}\right)_3$, and $\left(h_{\text{ln1}}^{(j)}\right)_{3+i}$, and the first three rows of $W_{\text{key}}$ are zero vectors, we get that the key corresponding to symbol $j$ is given by

$$k_k^{(j)} = \left(\left(h_{\text{ln1}}^{(j)}\right)^\top W_{\text{key}} + b_{\text{key}}\right)_k = (d/4)^{1/2}(W_{\text{key}})_{3+i,k} = \begin{cases} (d/4)^{1/2} & \text{if } k = j \\ 0 & \text{otherwise.} \end{cases} \tag{41}$$

and the query corresponding to symbol $j$ is given by

$$q_k^{(j)} = \left(\left(h_{\text{ln1}}^{(j)}\right)^\top W_{\text{query}}\right)_k = (d/4)^{1/2}(W_{\text{query}})_{3+i,k} = \begin{cases} c_1 & \text{if } k \in \{j_{i+1}, k_{i+1}\} \\ 0 & \text{otherwise.} \end{cases} \tag{42}$$

We thus have that the key matrix $K$ (i.e., a matrix whose $j$-th row is the $j$-th key vector $k^{(j)}$) will be a $b \times d$ matrix whose first $b$ columns are equal to $(d/4)^{1/2}$ times the identity matrix and all the remaining elements are 0. Consequently, the attention weights that will be used for computing the hidden representation corresponding to the $i$-th symbol are given by

$$a^{(i)} = \text{Softmax}\left(\hat{a}^{(i)} \odot \text{Mask}\right), \tag{43}$$

where

$$\hat{a}_k^{(i)} = \frac{\left(q^{(i)}K^\top\right)_k}{\sqrt{d}} = \begin{cases} c_1/2 & \text{if } k \in \{j_i, k_i\} \\ 0 & \text{otherwise.} \end{cases} \tag{44}$$

Then, since

$$c_1 = 2\left\lceil \ln\left(\frac{(i-2)(1/2-\epsilon)}{2\epsilon}\right)\right\rceil \implies e^{c_1/2} > \frac{(i-2)(1/2-\epsilon)}{2\epsilon} \tag{45}$$

$$\implies (1 - 2(1/2-\epsilon))e^{c_1/2} > (i-2)(1/2-\epsilon) \tag{46}$$

$$\implies e^{c_1/2} > (1/2-\epsilon)2e^{c_1/2} + (i-2)(1/2-\epsilon) \tag{47}$$

$$\implies \frac{e^{c_1/2}}{2e^{c_1/2} + (i-2)} > (1/2-\epsilon), \tag{48}$$

we have that, for $k \in \{j_{i+1}, k_{i+1}\}$,

$$a_k^{(i)} = \frac{e^{\hat{a}_k^{(i)}}}{\sum_{j=1}^b e^{\hat{a}_j^{(i)}}} = \frac{e^{c_1/2}}{2e^{c_1/2} + 1\cdot(i-2) + 0\cdot(b-i)} > 1/2 - \epsilon. \tag{49}$$

This shows that that the hidden representation corresponding to symbol $i$ after the attention module can be made arbitrarily close to the average of the value vectors corresponding to symbols $j_{i+1}$ and $k_{i+1}$.

Since the first two elements of $h_{\text{emb}}^{(j)}$ are $\text{sign}(s_j)$ and $-\text{sign}(s_j)$, we get that the value vector corresponding to symbol $j$ is given by

$$v_k^{(j)} = (h_{\text{ln1}}^\top W_{\text{value}} + b_{\text{value}})_k = \begin{cases} \text{sign}(s_j) & \text{if } k = 4 \\ -\text{sign}(s_j) & \text{if } k = 5 \\ 0 & \text{otherwise.} \end{cases} \tag{50}$$

The hidden representation corresponding to symbol $i$ after the attention module is then given by

$$h_{\text{att}}^{(i)} = \sum_{j=1}^{b} a_j^{(i)} v^{(j)} = a_{j_{i+1}}^{(i)} v^{(j_{i+1})} + a_{k_{i+1}}^{(i)} v^{(k_{i+1})} + \sum_{j \notin \{j_{i+1}, k_{i+1}\}} a_j^{(i)} v^{(j)}, \tag{51}$$

and we have that

$$s_{i+1} = 1 \implies (s_{j_{i+1}}, s_{k_{i+1}}) = (0, 0) \tag{52}$$

$$\implies v_4^{j_{i+1}} = v_4^{k_{i+1}} = -1 \tag{53}$$

$$\implies (h_{\text{att}}^{(i)})_4 \in [-1, (1 - 2\epsilon)(-1) + (2\epsilon)(1)], \tag{54}$$

$$s_{i+1} = 0 \iff (s_{j_{i+1}}, s_{k_{i+1}}) \neq (0, 0) \tag{55}$$

$$\implies v_4^{j_{i+1}} \in \{0, 1\}, v_4^{k_{i+1}} \in \{0, 1\} \tag{56}$$

$$\implies (h_{\text{att}}^{(i)})_4 \in [(1 - 2\epsilon)(0) + (2\epsilon)(-1), 1], \tag{57}$$

that is,

$$\left(h_{\text{att}}^{(i)}\right)_4 \in \begin{cases} [-1, -1 + 4\epsilon] & \text{if } s_{i+1} = 1 \\ [-2\epsilon, 1] & \text{if } s_{i+1} = 0, \end{cases} \tag{58}$$

with $(h_{\text{att}}^i)_5 = -(h_{\text{att}}^i)_4$, and $(h_{\text{att}}^i)_j = 0$ for all $j \notin \{4, 5\}$.

The hidden representation corresponding to symbol $i$ after the first residual connection is given by

$$h_{\text{res1}}^{(i)} = h_{\text{att}}^{(i)} + h_{\text{emb}}^{(i)}. \tag{59}$$

Since $(h_{\text{att}}^i)_5 = -(h_{\text{att}}^i)_4$ and all other entries of $h_{\text{att}}^{(i)}$ are zero, and $h_{\text{emb}}^{(i)}$ has a mean of 0, we have that

$$\text{mean}\left(h_{\text{res1}}^{(i)}\right) = \text{mean}\left(h_{\text{att}}^{(i)}\right) + \text{mean}\left(h_{\text{emb}}^{(i)}\right) = 0 + 0 = 0. \tag{60}$$

Also, since $h_{\text{att}}^{(i)}$ can contain non-zero entries,

$$\text{std}\left(h_{\text{res1}}^{(i)}\right) \geq \text{std}\left(h_{\text{emb}}^{(i)}\right) = (d/4)^{-1/2} \tag{61}$$

and since the entries of $h_{\text{res1}}^{(i)}$ cannot exceed $(d/4)^{1/2}$,

$$\text{std}\left(h_{\text{res1}}^{(i)}\right) \leq \sqrt{\frac{1}{d} \sum_{k=1}^{d} \left((d/4)^{1/2} - 0\right)^2} = (d/4)^{1/2}. \tag{62}$$

Then, since

$$h_{\text{ln2}}^{(i)} = \text{LayerNorm}_{\gamma_{\text{ln2}}, \beta_{\text{ln2}}}\left(h_{\text{res1}}^{(i)}\right), \tag{63}$$

we have that

$$(d/4)^{-1/2}\left(h_{\text{res1}}^{(i)}\right)_j \leq \left(h_{\text{ln2}}^{(i)}\right)_j \leq (d/4)^{1/2}\left(h_{\text{res1}}^{(i)}\right)_j \tag{64}$$

for all $j \in \{1, 2, \ldots, d\}$, which implies that

$$\left(h_{\text{ln2}}^{(i)}\right)_4 \in \begin{cases} [-(d/4)^{1/2}, (d/4)^{-1/2}(-1 + 4\epsilon)] & \text{if } s_{i+1} = 1 \\ [(d/4)^{1/2}(-2\epsilon), (d/4)^{1/2}] & \text{if } s_{i+1} = 0. \end{cases} \tag{65}$$

Since $(d/4)^{1/2} > 1$, it follows that

$$\left(h_{\text{ln2}}^{(i)}\right)_4 \in \begin{cases} [-d, (d/4)^{-1/2}(-1 + 4\epsilon)] & \text{if } s_{i+1} = 1 \\ [-2d\epsilon, d] & \text{if } s_{i+1} = 0. \end{cases} \tag{66}$$

Using the definition of $W_{\text{ff1}}$ and $b_{\text{ff1}}$ and the fact that $(d/4)^{1/2} > 1$, we get that

$$\left(h_{\text{ln2}}^{(i)} W_{\text{ff1}} + b_{\text{ff1}}\right)_6 = -(d/4)^{1/2}\left(h_{\text{ln2}}^{(i)}\right)_4 - 2d^2\epsilon \tag{67}$$

$$\in \begin{cases} [1 - 4\epsilon - 2d^2\epsilon, d^2] & \text{if } s_{i+1} = 1 \\ [-d^2 - 2d^2\epsilon, 0] & \text{if } s_{i+1} = 0, \end{cases} \tag{68}$$

and

$$\left( h_{\ln 2}^{(i)} W_{\text{ff1}} + b_{\text{ff1}} \right)_7 = (d/4)^{1/2} \left( h_{\ln 2}^{(i)} \right)_4 + 1 - 4\epsilon \tag{69}$$

$$\in \begin{cases} [-d^2, 0] & \text{if } s_{i+1} = 1 \\ [1 - 4\epsilon - 2d^2\epsilon, d^2 + 1 - 4\epsilon] & \text{if } s_{i+1} = 0. \end{cases} \tag{70}$$

Since $\epsilon = \frac{1}{5+2d^2} < \frac{1}{4+2d^2}$, we have that $\delta = 1 - 4\epsilon + 2d^2\epsilon > 0$, and thus

$$\left( h_{\text{ff1}}^{(i)} \right)_6 = \text{ReLU} \left( h_{\ln 2}^{(i)} W_{\text{ff1}} + b_{\text{ff1}} \right)_6 \in \begin{cases} [\delta, 2d^2] & \text{if } s_{i+1} = 1 \\ [0, 0] & \text{if } s_{i+1} = 0 \end{cases} \tag{71}$$

and

$$\left( h_{\text{ff1}}^{(i)} \right)_7 = \text{ReLU} \left( h_{\ln 2}^{(i)} W_{\text{ff1}} + b_{\text{ff1}} \right)_7 \in \begin{cases} [0, 0] & \text{if } s_{i+1} = 1 \\ [\delta, 2d^2] & \text{if } s_{i+1} = 0. \end{cases} \tag{72}$$

Then, using the definition of $W_{\text{ff2}}$ and $b_{\text{ff2}}$, we get that

$$\left( h_{\text{ff2}}^{(i)} \right)_6 = \text{ReLU} \left( h_{\text{ff1}}^{(i)} W_{\text{ff2}} + b_{\text{ff2}} \right)_6 = \left( h_{\text{ff1}}^{(i)} \right)_1 \in \begin{cases} [\delta, 2d^2] & \text{if } s_{i+1} = 1 \\ [0, 0] & \text{if } s_{i+1} = 0, \end{cases} \tag{73}$$

$$\left( h_{\text{ff2}}^{(i)} \right)_7 = \text{ReLU} \left( h_{\text{ff1}}^{(i)} W_{\text{ff2}} + b_{\text{ff2}} \right)_7 = \left( h_{\text{ff1}}^{(i)} \right)_2 \in \begin{cases} [0, 0] & \text{if } s_{i+1} = 1 \\ [\delta, 2d^2] & \text{if } s_{i+1} = 0, \end{cases} \tag{74}$$

while $\left( h_{\text{ff2}}^{(i)} \right)_8 = - \left( h_{\text{ff2}}^{(i)} \right)_6$, $\left( h_{\text{ff2}}^{(i)} \right)_9 = - \left( h_{\text{ff2}}^{(i)} \right)_7$, and $\left( h_{\text{ff2}}^{(i)} \right)_j = 0$ for all $j \in \{10, 11, \dots, d\}$.
Since

$$h_{\text{res2}}^{(i)} = h_{\text{ff2}}^{(i)} + h_{\text{res1}}^{(i)}, \tag{75}$$

we have that

$$\text{mean} \left( h_{\text{res2}}^{(i)} \right) = \text{mean} \left( h_{\text{ff2}}^{(i)} \right) + \text{mean} \left( h_{\text{att}}^{(i)} \right) = 0 + 0 = 0. \tag{76}$$

Also, since the entries of $h_{\text{res2}}^{(i)}$ cannot exceed $d^2$,

$$\text{std} \left( h_{\text{res2}}^{(i)} \right) \le \sqrt{\frac{1}{d} \sum_{k=1}^{d} (d^2 - 0)^2} = d^2. \tag{77}$$

Then, since

$$h_{\ln 3}^{(i)} = \text{LayerNorm}_{\gamma_{\ln 3}, \beta_{\ln 3}} \left( h_{\text{res2}}^{(i)} \right), \tag{78}$$

we have that

$$\left( h_{\ln 2}^{(i)} \right)_j = \left( \text{std} \left( h_{\text{res2}}^{(i)} \right) \right)^{-1} \left( h_{\text{res2}}^{(i)} \right)_j \ge d^{-2} \left( h_{\text{res2}}^{(i)} \right)_j \tag{79}$$

for all $j \in \{1, 2, \dots, d\}$. It then follows that

$$s_{i+1} = 1 \implies \left( h_{\ln 2}^{(i)} \right)_6 \ge \delta d^{-2} \text{ and } \left( h_{\ln 2}^{(i)} \right)_7 = 0, \tag{80}$$

$$s_{i+1} = 0 \implies \left( h_{\ln 2}^{(i)} \right)_6 = 0 \text{ and } \left( h_{\ln 2}^{(i)} \right)_7 \ge \delta d^{-2}. \tag{81}$$

Let

$$z = h_{\ln 3}^{(i)} W_{\text{out}}. \tag{82}$$

Then, using the definition of $W_{\text{out}}$ and the fact that $c_2 = \left\lceil \ln \left( \frac{1-\epsilon^*}{\epsilon^*} \right) \right\rceil \ge \ln \left( \frac{1-\epsilon^*}{\epsilon^*} \right)$, we get that

$$s_{i+1} = 1 \implies z_1 = \left( h_{\ln 2}^{(i)} \right)_7 (\delta^{-1} d^2 c_2) = 0 \text{ and } z_2 = \left( h_{\ln 2}^{(i)} \right)_6 (\delta^{-1} d^2 c_2) \ge c_2, \tag{83}$$

$$\implies e^{z_2} \ge \frac{1-\epsilon^*}{\epsilon^*} \tag{84}$$

$$\implies e^{z_2} - e^{z_2} + e^{z_2}\epsilon^* > 1 - \epsilon^* \tag{85}$$

$$\implies e^{z_2} > e^{z_2} - e^{z_2}\epsilon^* + 1 - \epsilon^* \tag{86}$$

$$\implies P_\theta(s_{i+1} = 1) = (\text{Softmax}(z))_2 = \frac{e^{z_2}}{e^{z_1} + e^{z_2}} = \frac{e^{z_2}}{e^{z_2} + 1} > 1 - \epsilon^*, \tag{87}$$

and

$$s_{i+1} = 0 \implies z_1 = \left(h^{(i)}_{\ln 2}\right)_7 \left(\delta^{-1} d^2 c_2\right) \geq c_2 \text{ and } z_2 = \left(h^{(i)}_{\ln 2}\right)_6 \left(\delta^{-1} d^2 c_2\right) = 0 \tag{88}$$

$$\implies e^{z_1} \geq \frac{1 - \epsilon^*}{\epsilon^*} \tag{89}$$

$$\implies e^{z_1} - e^{z_1} + e^{z_1} \epsilon^* > 1 - \epsilon^* \tag{90}$$

$$\implies e^{z_1} > e^{z_1} - e^{z_1} \epsilon^* + 1 - \epsilon^* \tag{91}$$

$$\implies P_\theta(s_{i+1} = 1) = (\text{Softmax}(z))_2 = \frac{e^{z_1}}{e^{z_1} + e^{z_2}} = \frac{e^{z_1}}{e^{z_1} + 1} > 1 - \epsilon^*. \tag{92}$$

We conclude that, for each $i \geq n$, $P_\theta(s_{i+1} = \text{NOR}(s_{j_{i+1}}, s_{k_{i+1}})) > 1 - \epsilon^*$.

