# OpenReview forum: "Guiding Transformers to Process in Steps"
_ICLR.cc/2022/Conference — ICLR 2022 Submitted_

### Official Review · Reviewer_eRnm · 2021-10-27

**Correctness:** 3
**Technical Novelty And Significance:** 2
**Empirical Novelty And Significance:** 2
**Recommendation:** 3
**Confidence:** 5

**Main Review:**

Strengths
=======
* I found this work interesting and it is important that researchers explore the limits of what can be learned with current deep learning methods.
* The text well-written.
* The authors provide the code in the supplementary material.

Weaknesses
==========

* In its current state, the work is too shallow for a conference paper. **Here are some additional questions** you could try to answer in your work to increase its depth / insightfulness:
  1. Intermediate computations seem to help for binary addition, what about other computations? Is there any task where the model would fail even when having access to intermediate steps?
  2. The study focuses on transformers but it is clearly applicable to other models such as LSTMs or DNCs. Why not trying with more architectures?
  3. Experiments are done with shallow and pre-trained transformers but it also would be interesting to see what is required to make a transformer learn to perform algorithmic tasks from scratch (how many layers, attention heads, amount of data, amount of supervision) and to compare it with your current models.
  4. It would be interesting to find an algorithmic task for which you can gradually choose the amount of intermediate guidance that you provide and see how much guidance do different kinds of models require.

* The system 1 / system 2 motivation is interesting but maybe there is too much emphasis on it given that the work is more about transformers than human cognition.

**Summary Of The Paper:**

The authors show that shallow and frozen pre-trained transformers can learn to perform algorithmic tasks if properly guided with intermediate computations. They experiment on binary bit addition, showing that including information about carry bits results in models that generalize significantly more than those trained with weaker supervision.

**Summary Of The Review:**

The main idea of this submission is interesting but the contributions are not significant enough for ICLR. In "Weaknesses" I provide some additional questions that the authors could address to increase the amount of contribution / insights.  For the moment I recommend to reject this work but I strongly encourage the authors to continue working on it!

---

### Official Review · Reviewer_urkD · 2021-10-31

**Correctness:** 2
**Technical Novelty And Significance:** 2
**Empirical Novelty And Significance:** 2
**Recommendation:** 3
**Confidence:** 3

**Main Review:**

# Summary
The paper presents the authors’ hypothesis: intermediate steps are necessary for Transformers to perform well on algorithmic tasks. They analyse Transformers on a simple binary addition task with and without intermediate steps. They show that only the one with immediate steps and supervision over these steps can generalise. In contrast to most of the prior work, the authors argue for supervision in the output domain. This makes the approach architecture-agnostic.

# Strong points
-   The topic is interesting and relevant
-   The paper is clear and easy to understand
-   The supervision is applied as additional outputs, which makes the approach architecture agnostic

# Weaknesses
-   The 2nd implication (Training) in section 3 is not generally true. There might be many reasons for the NN not learning a task, for which strong supervision cannot help. For example, one can just use a state size of 1 or set initial weights to zero.
-   The Composability implication of section 3 is not generally true. Having atomic operations is just a necessary but not sufficient condition for building more complicated algorithms from them. Decomposing novel problems to the known atomic operations is a very hard, unsolved problem.
-   The authors appear to mix two things: the availability of extra computation steps and the strong supervision of the intermediate outputs. Because of this, it is unclear whether their findings show that additional computation steps or additional supervision are needed.
-   The authors claim in section 5 that their results in table 1 prove 2nd implication from Section 3. However, showing that it works in a particular case does not prove the general statement.
-   The claim of section 3, that “no matter how many atomic operations separate outputs from inputs…” is true only if we ignore the additional complexity of decomposing the problem and combining the intermediate results, or if this has a fixed complexity, which may or may not be the case.
-   The authors seem to define the intermediate steps purely in the output space. But in fact, intermediate results can be computed in different layers of the network, and even with extra output steps, this might be enough for solving the problem.

# Questions/suggestions
-   The weaknesses from above have to be fixed
-   The experiments either have intermediate steps with supervision, or they don’t have intermediate steps at all. This makes it unclear whether the performance gain comes from the additional computation steps or additional supervision. A more proper way to evaluate this would be to add a third version, which has the same number of output steps as the current strong supervision variant, but with no supervision applied to the intermediate steps.
-   For adaptive computation, please add citations to [1] and maybe [2]
-   For Veličković’s work in the references, the authors only mention a work that is concerned about graph-structured problems. However, his recent work is more general focuses on diverse types of data. On the other hand, Transformers can also be understood as graph networks with a fully connected graph (which is also common in the GNN literature)
-   There is a mistake in eq (7) in the appendix: as defined here, if the logits are negative, the attention score will become nan (inf/inf). The mask should be additive, and the “remaining elements” should be 0 instead of 1. Of course, there are other ways to achieve the same result, but the one described in the paper is incorrect and can result in trainability issues.

# Other
I have not verified the correctness of the hardcoded weights from the Appendix.

# Citations
[1] Jürgen Schmidhuber: Self-Delimiting Neural Networks (2012)
[2] Banino et al: PonderNet: Learning to Ponder (2021)

**Summary Of The Paper:**

The paper presents the authors’ hypothesis: intermediate steps are necessary for Transformers to perform well on algorithmic tasks. They analyse Transformers on a simple binary addition task with and without intermediate steps. They show that only the one with immediate steps and supervision over these steps can generalise. In contrast to most of the prior work, the authors argue for supervision in the output domain. This makes the approach architecture-agnostic.

**Summary Of The Review:**

As for the weaknesses, I have to recommend the rejection of the paper. There are flaws in multiple claims, and the authors seem to mix the availability of computation steps and supervision of them.

---

### Official Review · Reviewer_4ZwD · 2021-11-02

**Correctness:** 3
**Technical Novelty And Significance:** 2
**Empirical Novelty And Significance:** 2
**Recommendation:** 3
**Confidence:** 4

**Main Review:**

I find both the motivating question ('how to get deep learning models to carry out processes associated with system 2?'), and the hypothesis ('by providing intermediate results as supervision') to be generally compelling. Humans are clearly provided with such supervision through the course of formal eduction (e.g. by being a given a demonstration for how to solve a multi-digit arithmetic problem), so this seems like a promising direction to pursue. In its current state however, the paper suffers from two significant limitations:

1. All experiments are performed on the very simple and limited domain of binary addition. This seems like a good testbed for trying out some initial ideas and getting a handle on the problem, but it would be good to show that the method is also useful in richer domains, e.g. more complicated mathematical reasoning problems such as those in the dataset from [1] (notably, problems requiring a number of intermediate steps seemed to be particularly challenging for the models tested in that paper, so it is plausible to think that the presently proposed method might be more useful there). With experiments only performed on such a simple domain, it is very difficult to draw any general conclusions from these results.
2. The paper's primary comparison is between a 'weak supervision' condition and a 'strong supervision' condition. There are two major differences between these conditions, and thus there is currently a confound concerning which of these differences is responsible for the differences in performance. The first major difference is that, in the 'weak supervision' condition, the model is expected to produce an answer in a single forward pass, whereas in the 'strong supervision' condition, the model produces intermediate results which are then fed back into the model as input. The second major difference is that, in the 'strong supervision' condition, the model is furthermore provided with supervision telling it what the intermediate results should be. An important control model is one that produces a final result via a series of intermediate results (as in the 'strong supervision' condition), but in which there is no supervision for what those intermediate results ought to be (as in the 'weak supervision' condition). The central proposal of the paper is that supervision is necessary to allow the learning of useful intermediate computations, but, without this control condition, it is impossible to know whether the 'strong supervision' model succeeds because of this supervision, or simply because it is allowed to compute intermediate results.

Other issues:
- Given the simplicity of the task, I wonder to what extent the pre-trained transformer is actually necessary for the results in section 6 ('composability'), as opposed to the fine-tuned input and output layers doing most of the work. A good control would be to test how the same model would perform when the 'body' of the model is just randomly initialized (and frozen), as opposed to being pre-trained.
- Are all results presented for just a single model instance, or averaged over multiple model instances? Ideally, these results would be averaged over multiple model instances, with the number of model instances, and some measure of variance, reported.
- The authors might consider citing, and discussing this paper [2], in which recurrent networks are used to generalize from simple to more complex computations via reuse of those simpler computations (notably though, this is *without* supervision on the intermediate results).
- There is not adequate explanation of the notation used in the argument presented in section 4 (e.g. the indices *n* and *l* are not explained until section 5).
- In the 'related work' section, 'Mirman et al. (2018)' is mentioned twice in the same sentence, in a manner that makes it sound like these should be referring to two separate papers. Is this a mistake?

[1] Saxton, D., Grefenstette, E., Hill, F., & Kohli, P. (2019). Analysing mathematical reasoning abilities of neural models. arXiv preprint arXiv:1904.01557.

[2] Schwarzschild, A., Borgnia, E., Gupta, A., Huang, F., Vishkin, U., Goldblum, M., & Goldstein, T. (2021). Can You Learn an Algorithm? Generalizing from Easy to Hard Problems with Recurrent Networks. arXiv preprint arXiv:2106.04537.

**Summary Of The Paper:**

This paper proposes that, in order to enable deep learning systems to carry out processes akin to 'system 2' (a term from cognitive psychology, referring to deliberate, step-by-step reasoning processes), these systems should be trained through supervised learning to perform complex computations via a series of simpler, intermediate computations (i.e. by providing the results of those intermediate computations as a source of supervision). Experiments are focused on the domain of binary addition, where it is shown that 1-layer transformers are capable of performing the task for very long bit strings when allowed to do so through a series of intermediate operations, but are limited to very short bit strings when forced to perform the task in a single forward pass.

**Summary Of The Review:**

The proposed approach is promising, but the experiments are only performed on a very simple, limited task domain, and an important control experiment is missing.

---

### Official Review · Reviewer_KKJC · 2021-11-06

**Correctness:** 3
**Technical Novelty And Significance:** 2
**Empirical Novelty And Significance:** 1
**Recommendation:** 3
**Confidence:** 4

**Details Of Ethics Concerns:**

Don't see ethical concerns.

**Main Review:**

While I value the high-level goal of the paper and think it is important to explore and establish the necessity of sequential reasoning for system 2 tasks, I feel that in the current form of the paper it doesn’t explore it rigorously or extensively enough to corroborate its claims in a solid manner:
* **Too narrow task**: The focus on very specific addition problems in a quite particular non-intuitive input-output format doesn’t support the generality of the claims about the necessity  of guiding models through intermediate supervision to make them capable of tackling algorithmic tasks. There could be a much richer variety of algorithmic tasks that fit the domain of the paper and that I would hope to see when I started to read it, that could be explored to make the claims better supported - graph traversal, sorting, reasoning over sequences, summing decimal numbers in different formats (e.g. input x+y+z.. and outputting the result), etc etc. It feels to me that the current focus narrows down the paper preventing it from substantiating such general claims.
* **Extending experiments**: assuming focusing on the particular addition task in the paper, I think it could still benefit from a larger variety of experiments. Currently there are only a handful of experimental results, basically showing that strong-supervision makes 2 models (pre-trained and from scratch) achieve perfect 1.0 accuracy on the addition task, and fail otherwise. Further exploration into (1) the size of training data, (2) the amount of supervision (interpolation between just weak and strong), (3) the complexity of the addition task in different dimensions, and (4) the transformer configuration (number of parameters, heads, layers) impact performance would be useful to get more intuition on what are the reasons behind models’ failure or success in different settings.
* **Choice of tested model**: The paper mainly focuses on 1-layer 1-head transformer, but doesn’t justify the choice of the model. It could be that even though theoretically such a model can represent any finite function, it’s too constrained for allowing it to figure out good solution for the task through training without intermediate supervision, but given that the chosen model is very small and limited to begin  that doesn’t necessarily prove that intermediate supervision is necessary for models. Indeed, naturally it is generally true that providing more supervision to tasks would make them simpler, but if a reasonable increase in model’s capacity (more layers, more heads) could allow avoiding that extra supervision, then why is it still necessary? I think it could be really interesting to show the necessity of intermediate supervision but I think that will be better demonstrated by finding a case where even if you add more layers, heads, or parameters to the model, it still fundamentally doesn’t manage to solve an algorithmic task, unless the intermediate supervision is given. I think tasks that have exponential nature in terms of the search space of potential solutions could be good candidates to show that since their space may quickly outgrow the capacity of a model, even if increased in size or complexity.
* **Exploration of other classes of models**: I think for that paper in particular that focuses on sequential reasoning and the necessity of intermediate steps supervision, it would be very natural to also explore recurrent approaches (RNNs, LSTMs) and their performance and compare it to that of transformers.
* **Discussion about systems 1 and 2**: The discussion about systems 1 and 2 feels a bit honestly hand-wavy and not precise enough -- given the extent to which these terms are used and discussed here, it would be good to start by some clear definitions in the context of the paper: how to you define system 1 tasks and systems 2 tasks here? Are all math problems considered system 2? Are “easy-enough” to remember math problems should be considered system 1? If a model is pre-trained on a task does it make it system 1 or 2 task (because if the model already is skilled in answering that particular sort of question one might argue it doesn’t require explicit sequential reasoning anymore turning it into system 2).
* **Motivation section**: I feel that the motivation section in its current form is not compelling and doesn’t answer key questions about the choice of tasks and the design of the experiments discussed above. I think it may be better to work more on the presentation of the paper.


**Summary Of The Paper:**

The paper’s main claim is that sequential reasoning is necessary for system 2 tasks, and exploring addition as an example for that, showing that a 1-layer 1-head transformer fails to compute the sum of numbers directly (although theoretically shown to be able to compute any finite function)  but does work when being supervised by intermediate results. Then they show that a pretrained transformer likewise fails withouts intermediate supervision on a binary addition task.

Update: I do apreaciate the author's revision but since no response has been provided to my comments and/or was addressed by the revision, I unfortunately keep my score.

**Summary Of The Review:**

To summarize my review, I think that while the high-level goal of the paper of exploring the necessity of intermediate supervision is interesting and important, the experimental evaluation is not rigorous enough to demonstrate it, missing along multiple key dimensions, which include:
* Choosing a task that is too narrow
* Very narrow exploration of models and experimental settings within the chosen task
* Unjustified/unexplained choice of an unnaturally constrained model (1-layer 1-head transformer)
* Lack of exploration of other relevant models (e.g. recurrent networks)

For these reasons I unfortunately recommend rejecting the paper in its current form, but do encourage the authors to extend and explore it further in order to improve it!

---

### Author Response · Authors · 2021-11-23
**Revision summary**

We would like to thank all of the reviewers for their detailed reviews. We respect the criticism and we appreciate the words of encouragement. The following are the main changes we have made to the paper in response to the reviewers’ comments:

* We added results for the “medium supervision” case where the model generates intermediate results but is only trained to predict the outputs (i.e., no loss is computed on the intermediates);
* We included additional figures and comments providing more context on the impact of hyperparameters on the model’s performance in training experiments;
* We added justifications for the choice of binary addition as the target task and the choice of a 1-layer 1-head Transformer as the target model;
* We modified the main text to be more careful and precise when drawing conclusions from our results (e.g., emphasizing that our empirical demonstrations only serve as evidence for rather than proof of the hypotheses);
* We fixed the error in the definition of the Transformer in Appendix A pointed out by Reviewer urkD.

---

> ### Comment · Reviewer_urkD · 2021-11-25
> **Response for the revised version**
>
> I am happy to see the changes made by the authors, especially since they toned down the "proving" and "should" parts.
>
> The new experiments with "Medium supervision" could have the potential to prove that supervision is necessary for good performance. However, these experiments have a fatal flaw in their current form, which explains why the new setting performs so badly compared to even the "weak supervision" case. Namely, during the training, it uses teacher forcing: the "ground truth" intermediates are fed to the network in every step. The network can learn to use these intermediates easily during the training. However, since there is no supervision on the intermediate outputs of the network, it is almost sure that it will not learn to generate them, so during the testing, the network will fail miserably (as we see). A better approach would be not to use teacher forcing in this case (use it only if the intermediate steps are supervised). This would force the network to do most of the processing through the internal attention layers (it could also figure out to use its own noisy output, which is fed back, although I find that unlikely). Furthermore, this would make the training and testing regimes similar, which is not currently the case.

---

> ### Comment · Reviewer_4ZwD · 2021-11-27
> **Reply**
>
> Thanks to the authors for these updates, particularly the new 'medium supervision' condition. In principle I think this condition could support the claim about the necessity of supervision on the intermediate outputs, but I have two concerns:
> 1) just confirming that accuracy for the medium supervision condition is not based on also getting the intermediate outputs correct? I believe this is the case based on the wording in the paper, but I'm just confirming since I think that would obviously be an unreasonable evaluation metric given the lack of explicit training on those outputs (i.e. the question should be only whether those additional steps allow the model to get the final answer correct).
> 2) I agree with reviewer urkD, it would be better to eliminate teacher-forcing in the training of the medium supervision model, since that will introduce a significant difference between training and test.
>
> On the whole, I am still very unconvinced that binary addition is adequate as the sole empirical test of the model. I see that it has advantages in terms of elucidating the relevant issues, but this needs to be combined with experiments on a wider range of tasks to validate that the idea is not specific to this domain.

---

> > ### Comment · Reviewer_urkD · 2021-11-27
> > **Clarification**
> >
> > I just want to clarify: My concerns are not about 1. As far as I understand, the authors measure performance just based on the final output, not the intermediate output, which is correct. However, since the model is always trained with the correct intermediates fed to its inputs (teacher forcing) but not trained to produce these on its own, it is expected to fail at test time. No teacher is forcing is applied in test time: the fed back values will be untrained, thus not the correct sequence of intermediates. Thus, this setup does not tell us whether the extra steps or the supervision is the key factor. If one would not use teacher forcing, the model would be forced to pass information between steps by other (differentiable) means that are trained, like attention, which could be still sufficient to perform the task.

---

> ### Comment · Reviewer_eRnm · 2021-11-29
> **Reply**
>
> Thanks for the updates!
>
> I value your effort, it has certainly improved the submission. However, the rebuttal only focuses in some of the aspects mentioned by the reviewers (clarifications, medium supervision...) while not addressing many points such as de ones pointed by KKJC (narrow task, extending experiments, exploration of other classes of models) and the rest of the reviewers.
>
> Thus, although it has great potential, I still consider that the submission is within the "reject" region.

---

### Author Response · Authors · 2021-11-30
**Response to the feedback on the revision**

Thanks to all of the reviewers for the additional feedback. We do agree with the concerns about the current implementation of the “medium supervision” experiments raised by Reviewer urkD. However, if our understanding of the proposed alternative is correct, not using teacher forcing during training would lead to non-standard gradient propagation which might be problematic. If we decide to use the model’s predictions of the intermediates instead of the “true” ones during training, we need to autoregressively generate them one at a time. Then, when predicting the actual outputs, we would have to propagate gradient back throughout this whole autoregressive generation (which might consist of tens or hundreds of steps), instead of treating the intermediates as inputs like in the conventional case. This can be implemented (and we will include such experiments in the future iterations of the paper), but due to the unconventional nature of the resulting training regime, it is not clear whether this would allow us to draw strong conclusions.

As to question 1 from Reviewer 4ZwD, you are correct — the accuracy is based only on the symbols representing the sum; the intermediate outputs are not taken into account.

---

> ### Comment · Reviewer_urkD · 2021-11-30
> **Response**
>
> We agree with the authors that the setup without full supervision is more challenging than the one with it. However, the current setup does not have even a theoretical chance of working (except in 2 cases, both extremely unlikely: 1. the network decides to ignore all of the fed back input, 2. the output layer will, by chance, generate the correct intermediate outputs). Thus, concluding anything based on this experiment is misleading.
>
> In my experience, the following setup usually works in such cases: train on dynamic length examples (more challenging than a fixed length, so can't compare directly) and use a mixture of short and long examples. The short examples typically provide sufficient gradients such that the network can learn.
>
> Overall, I think the paper is still not strong enough to recommend acceptance. However, I encourage the authors to improve it because the question is interesting and important.

---

### Decision · Program_Chairs · 2022-01-20

**Decision:**

Reject

**Comment:**

This paper offers new ideas about the key question of how to extend modern Transformer architectures to solve problems that require more reasoning steps than the model can implement in a single-step forward pass. Reviewers were unanimous that the problem is important, and that the paper is a step in a promising direction. However, reviewers were also unanimous that the proposed experiments are too narrow to be the basis for any confident new claims in this area and that, in addition, the experimental design has a confound that makes it difficult to interpret, even after the addition of a new condition during discussion.